# A Conflict-Detecting and Early-Warning System for Multi-Plan Integration in Small Cities and Towns Based on Cloud Service Platform

**Ningrui Du [1,*], Ming Zhang [2] , Jingnan Huang [1] and Guoen Wang [1]**

1    School of Urban Design, Wuhan University, Wuhan 430072, China
2    School of Architecture, The University of Texas at Austin, Austin, TX 78712-1160, USA
*    Correspondence: 00200149@whu.edu.cn

**Abstract:** Multi-plan integration (MPI) is a major effort initiated by China's State Council for the purpose of streamlining development plans made by various public agencies in provincial and city governments.  Small cities and towns are facing challenges to achieve MPI goals due to lack of technological infrastructure and professional expertise. This article presents a system to assist small cities and towns to carry out their MPI tasks. The system, named conflict-detecting and early-warning for MPI (CDEW4MPI) is developed based on a cloud service platform. CDEW4MPI enables small cities and towns in remote locations to detect inconsistency and conflicts among multiple plans. The system includes two modules.  One is conflict-detecting, which identifies spatial conflicts in boundary designation among different plans from different agencies.  The other is early-warning, which analyzes and reports potential encroachment of proposed local plans to urban growth boundary, the baseline for ecological protection, and the farmland under permanent preservation. CDEW4MPI was implemented as a demo project in Shennongjia Forestry District, a municipality in the western mountainous region of Hubei Province, China.  The paper presents the design of CDEW4MPI and its implementation in Shennongjia and draws lessons from the Shennongjia case for broad interests in smart management of spatial resources.

**Keywords:** multi-plan integration (MPI); spatial planning; conflict detecting; conflict early-warning; cloud service planform; Shennongjia forestry district

## 1. Introduction

China has a long and strong tradition of plan making for economic, social, and urban and rural development.  Nearly all public agencies make and update their plans on a regular basis. For example, national and local development and reform commissions make plans for economic and social development and project investments. The urban and rural planning agencies at all levels of governments take responsibility for planning for land use and spatial development for cities, towns, and villages. Land and resources management agencies at different levels prepare and implement plans for farmland protection and land supply for urban development. Departments of Ecology and Environment at different levels focus on ecological protection and pollution mitigation. More often than not, these plans exhibit inconsistencies or even conflicts between their agency-specific goals, objectives, standards, and zonal delineation of developments or controls. The inconsistencies and conflicts among the plans often cause confusions, inefficiency, and waste of resources when the plans are implemented [1–3].

As an effort to enhance inter-agency coordination and to improve planning governance and services, China's State Council initiated a program, multi-plan integration (MPI), to unite various

plans into one integrated plan system. The program was part of the effort to implement National New Urbanization Planning (2014), while the practical work pertaining to MPI has been ongoing since 2000 [4]. A number of provinces and large cities, for instance, Ningxia province, Hainan province, and cities of Guangzhou and Xiamen, carried out MPI pilot work [5–8]. Still, challenges facing MPI implementation remain strong, especially in small cities and towns. Data standards adapted by various agencies often vary, posing technical and institutional challenges to MPI. Most small cities and towns do not have sufficient capacities (technical, financial, and institutional) to collect and maintain the essential datasets and information needed for MPI and to carry out MPI tasks [2,9].

This paper presents an MPI support system targeting small cities and towns. Building on the databases available in provincial geo-information center via cloud-computing platform, our team developed a system of conflict-detecting and early-warning for multi-plan integration (CDEW4MPI). CDEW4MPI enables planning professionals and decision makers in small cities and towns to access the data resources and technical as well as professional knowhow available remotely in the provincial geo-information center to achieve MPI goals. A CDEW4MPI system was implemented as a demo project in Shennongjia Forestry District, a municipality in the western mountainous region of Hubei Province, China.

Following this introduction, the paper reviews related studies and presents an analysis of the characteristics of the multi-plan conflicts in China's planning context. The paper then introduces the system development and functions of CDEW4MPI. Next, the paper describes the relationship between CEDW4MPI and the cloud service platform, illustrated through the example of CEDW4MPI system application in the demonstration area, i.e., Shennongjia. Lastly, the paper discusses the lessons learned from the demo project and makes concluding remarks.

## 2. Review of Related Studies

Table 1 shows typical plans made by various agencies in Chinese cities, including small cities and towns. The plans concerning spatial issues mainly originate from four agencies, development and reform, urban and rural housing and construction, land management, and environmental protection. The departments responsible for making these plans follow their own traditionally authorized responsibilities and working programs. There exist inconsistencies among the plans in planning objectives, planning period, and planning contents. Concerning the issues of spatial governance, different plans abide by their own spatial regulations and adopt different functional zoning methods, which leads to inconsistent delimitation of spatial control boundaries and elements conflict of spatial management. Moreover, the data format and coordinate system used in the final planning results are discrepant and the land use classification standards also differ. The inconsistencies and conflicts among the plans from multiple agencies ultimately result in the chaotic situation of urban and rural construction management in the reality.

In small cities and towns, individual departments often have to correspond multi-level higher authorities and each plan must accord with its superior planning as guidance. Spatial inconsistency of multiple plans becomes amplified through the vertical transition from the upper level of plans down to the lower levels. Under the circumstance that each department does its own business and lacks well harmonized mechanisms in management, which make things worse, the difficulty of spatial coordination in urban and rural construction management of small cities and towns is increased. While MPI came with a great intention, achieving MPI is not an easy task as it requires significant technical, organizational, and financial commitments from a variety of institutional stakeholders.

**Table 1.** Differential analysis on spatial regulation of four kinds of planning in the small cities and towns.

| | Economic and Social Development Planning | Urban and Rural Planning | Land Use Planning | Environmental Protection Planning |
|---|---|---|---|---|
| Agency | Development and Reform | Urban and Rural Housing and Construction | Land Management | Environmental Protection |
| Planning period | 5 years | 20 years | 15 years | 5 years |
| Land use classification regulation | ‾‾‾ | Code for classification of urban land use and planning standards of development land (GB50137-2011) Code for town planning (GB50188–2007) | Current land use classification standards (GB/T21010–2007) Compiling rules for land use planning at country level (TD/T 1024–2010) | ‾‾‾ |
| Coordinate system | Non coordinate Sketch Map | GCS_Beijing_1954 or local coordination system | GCS_Xi'an_1980 coordination system | According to the base map |
| Functional Zoning | Urban area\Agriculture area\Ecological area | Residential area\Commercial area\Industrial area etc. | Urban constructional area\basic farmland protection zone\general farmland protection zone | Environmental functional zoning based on single element\Integrated Environmental functional zoning |
| Spatial governance zoning | Priority development zone\Key development zone\restricted development zone\Forbidden development zone | No construction zone\restricted construction area\proper construction area\built-up area | Forbidden construction zones\restricted construction zones\Conditional construction zones\Permitted construction zones | Core protected areas\Key protected areas\General protected areas\Development control areas |

Source: Compiled by the authors according to the planning regulations from corresponding ministries.

MPI and the establishment of a new spatial planning system in China have thus attracted great research interest, especially after 2013 [9]. Wang and Liu analyzed the multi-plan conflicts and actual dilemma based on the detailed exploration of the context and problems of spatial planning system in China [2]. Liu et al. indicated that the obstacles in MPI are discrepancy in technical standards and disharmony at the institutional level [10]. Increasingly, scholars have transferred their research interests in the fundamental reasons and mechanism of MPI for the reform of spatial planning system from different perspectives [11–13]. Hu and Zhou [11] stated that the key to MPI is the establishment of a system to coordinate the relations between the central and local government. Wu et al. [12] insisted that the technical problems for MPI should not be difficult due to the great progress of geographic information system science while the multi-plan combination reforms are not successful as result of empty institutions. Zhou et al. [13] tried to establish a coordinated spatial planning system of MPI and mentioned that the use of information and communication technologies (ICT) can be very useful for cooperation.

Institutional reform for establishing a new spatial planning system has been widely discussed in China. However, from the technical perspective, challenges remain strong for small cities and towns facing MPI implementation, especially on integrated information platform and technical standards. The coordination of planning techniques of different plans is still urgently needed in these areas. Chinese cities follow a quite rigid administrative hierarchy, which influences the allocation of resources, such as financial investment, public facilities, and various development opportunities [14,15]. The program of MPI in large cities and the cities in the developed areas could have access to the support of adequate funding from different sources and strong technical force and personnel. But most small cities and towns, especially in the remote areas, do not have the capacity to collect and maintain essential datasets and information. Local planning institutes have had difficulties in managing data resources to meet the MPI needs. In order to deal with the dilemma at the local level, some smart planning and management technology need to be applied to help small cities and towns reach adequate data and expertise in the advanced areas and large cities. Nowadays, the technology development makes it possible [16].

Cloud computing technologies have become one of the most significant scopes for IT innovation and is much more mature at present. The National Institute of Standard and Technologies (NIST) defined Cloud computing as "a model for enabling convenient, on-demand network access to a shared pool of configurable computing resources (e.g., networks, servers, storage, applications, and services) that can be rapidly provisioned and released with minimal management effort or service-provider interaction" [17]. It is "a new paradigm shift in which including computing resource services, soft application of distributed systems and data storage" [18]. Cloud service models can be grouped as software as a service (SaaS), platform as a service (PaaS), and infrastructure as a service (IaaS) [19,20]. Many scholars indicate that cloud computing can possess the advantages of low costs to host big data, provision of geographically distributed computing resources, internet-based communications to implement and provide services for users, and a shared pool of configurable resources (e.g., computing power, storage, networking, and applications) [21,22]. Therefore, it can greatly improve the quality of smart city services [23].

This study is part of a large research project on "Technology Integration and Demonstration of Intelligent Planning, Construction and Management in Small Cities Cluster", which was executed from 2015 to 2017. Building on the databases available in provincial geo-data center via cloud-computing platform, our team developed a system of conflict-detecting and early-warning for multi-plan integration (CDEW4MPI) targeting small cities and towns.

## 3. Research Approach and the Pilot Application

### 3.1. The Approach to CDEW4MPI System Development and System Functions

Figure 1 illustrates the design and development framework of the CDEW4MPI system. There are several steps, as following:

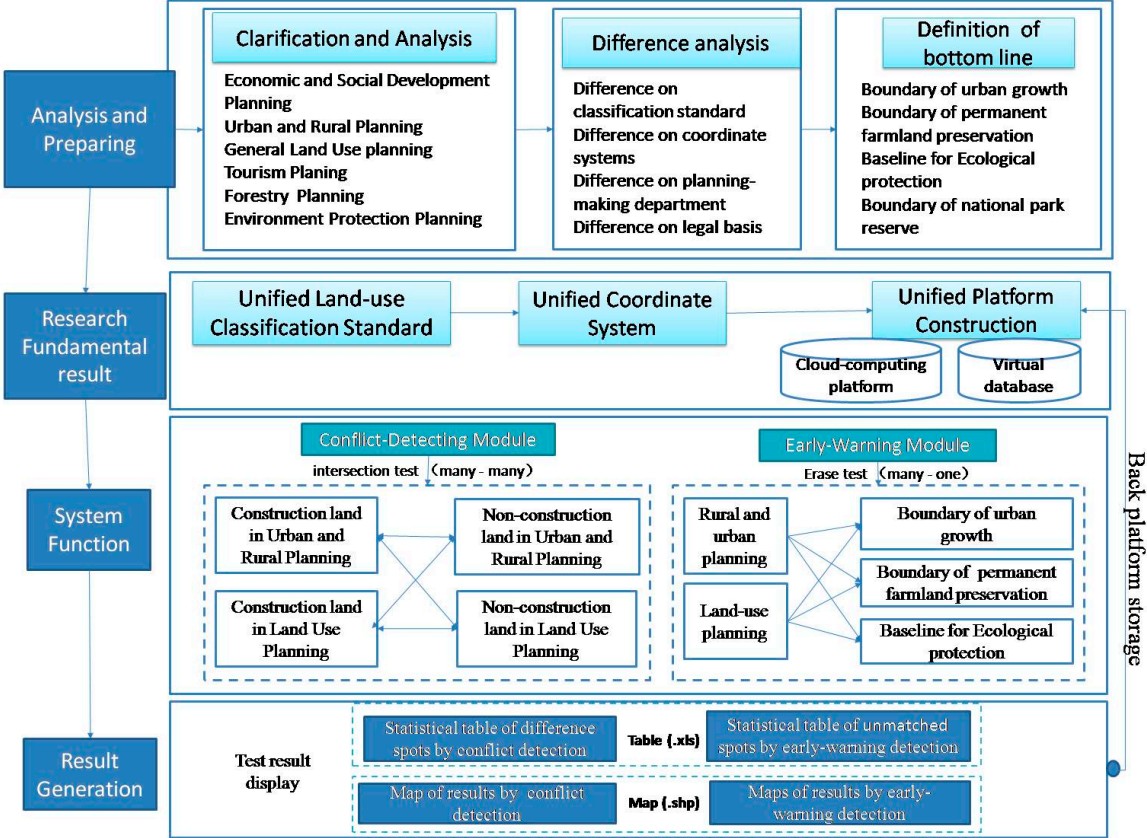

**Figure 1.** Framework of conflict-detecting and early-warning for multi-plan integration (CDEW4MPI) development.

At the initiation of the research, we analyzed the requirements and characteristics of small cities and towns, as well as the factors that generate inconsistencies inside different planning data from various spatial plans. There are four channels for us to collect information: First, in the demonstration area i.e., Shennongjia, we invite the key persons of many departments to attend the meeting, including development and reform commission, housing and construction bureau, land and resources bureau, forestry bureau, transportation bureau, statistical bureau, and the directors of eight towns, etc. During the meeting, we discussed with them the realistic dilemma, real requirements, and suggestions for MPI from their point of view to collect first-hand information. Second, we visited these departments to collect the second-hand data, especially the planning maps and materials. Third, our team conducted several field investigations to interview local people and indicate the specific problems and the areas with serious technical difficulties. Fourth, data from various other public resources were collected, including official government websites, portal websites of local governments, and university libraries.

The project team then developed a unified data classification scheme (e.g., land use classification system), digitalized and georeferenced data layers that initially came from non-digital or PDF sources, and constructed a database in a unified platform. This time-consuming but critical step of work is to unify land use classification and coordinate system of various plans. In all plans from different departments, urban and rural plans (master planning and detailed planning) from the housing and construction bureau and land-use plans from the land and resources bureau are central to MPI. The data types and data structure are technically complicated. Resolving the technical problems of conflict detection helps smoothen MPI with other plans. Considering application needs, the project team designed the functions of CDEW4MPI system to include two modules: Conflict-detecting module and early-warning module.

Finally, output of result is considered according to the basic requirements from the local government and the future management in a sustainable way. CDEW4MPI produces two kinds of reports: Discrepancy information sheet in Excel format and difference image spot in graphic format. All the results can be output to the local computers for printing and uploaded to the cloud service planform for storage management.

## 3.2. Integration of CDEW4MPI with Cloud-Computing Platform

CDEW4MPI system is based on the cloud service platform developed by Geo-Information Center in Hubei province, another team of the general research project. Their work makes it possible that the rich data sources are available in provincial geo-data centers and a cloud-computing platform is accessible and operable from remote locations. The integration of CDEW4MPI with the cloud-computing platform is shown in Figure 2.

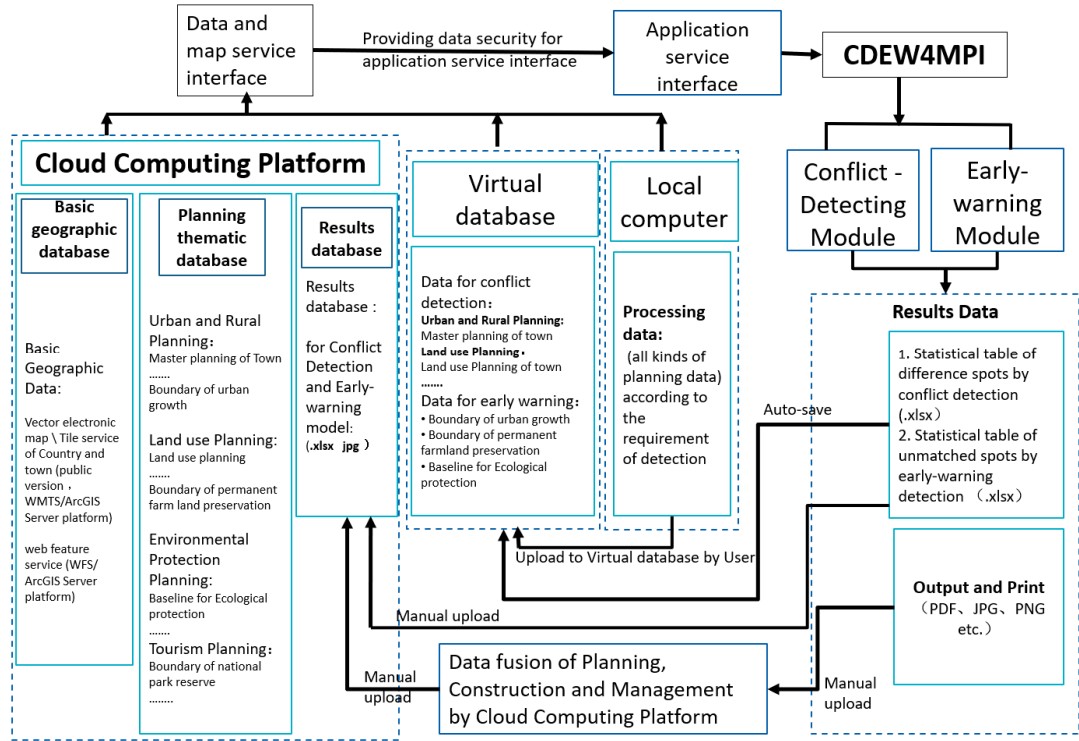

**Figure 2.** Integration scheme of CDEW4MPI with cloud-computing platform.

CDEW4MPI contains three technical layers: Platform, system core, and support. Plan-making data from various public agencies are stored in the cloud platform layer. The system core offers an interface for users to perform the main analysis functions. The support layer specifies network, hardware, and software environment needed for cloud-computing-based operations and analyses. The CDEW4MPI is developed on Asp.Net, ArcGIS API for JavaScript, and Asp.net for ArcGIS.

As shown in Figure 2, the basic geographic information database the planning and management information database is managed uniformly through the cloud service platform. CDEW4MPI can retrieve the data for operation and can also upload the updated standardized planning data to the cloud service planform. The application service interface of CDEW4MPI is integrated into the portal website of the cloud service planform to facilitate users' one-stop use of this system. All the results detected by CDEW4MPI, such as spatial conflicts and early-warning of breaking the bottom lines, can be uploaded to the planning results database on the cloud service planform through corresponding format.

### 3.3. An Overview of the Case Study Area as System Application Demonstration

Shennongjia Forestry District (SFD) was selected as the demonstration area of CDEW4MPI due to its typical characteristics. SFD is located in the western Hubei province (Figure 4), which is a unique municipal designation in China. It became an independent administrative unit approved by the State Council in 1970. By the end of 2018, SFD has six towns and two townships under its jurisdiction. Located in the western border of Hubei Province, it possesses a typical remote mountainous character. The total area is 3253 square kilometers with permanent population of 768,000 (but registered population of 78,900) at the end of 2017 [24]. Due to its vast forestry resources and the forestry area accounting for more than 85% of the whole region, it is also named 'forestry district' in China. Under the jurisdiction of Shennongjia Forestry District, the six towns and two townships are scattered across the mountainous topography and landform, each with its unique character.

The CDEW4MPI was implemented as a demo project in two towns of SFD: Songbai and Xinhua. Songbai is the municipal seat of Shennongjia, functioning as SFD's center of administration, culture and public services, and other supporting industries related to tourism. It is located northeast of SFD (Figure 3), with an area of 328 square kilometers and a total population of 29,860 in 2017. Being the capital town of SFD, Songbai offers a relatively strong capacity of planning, management, and technical infrastructure.

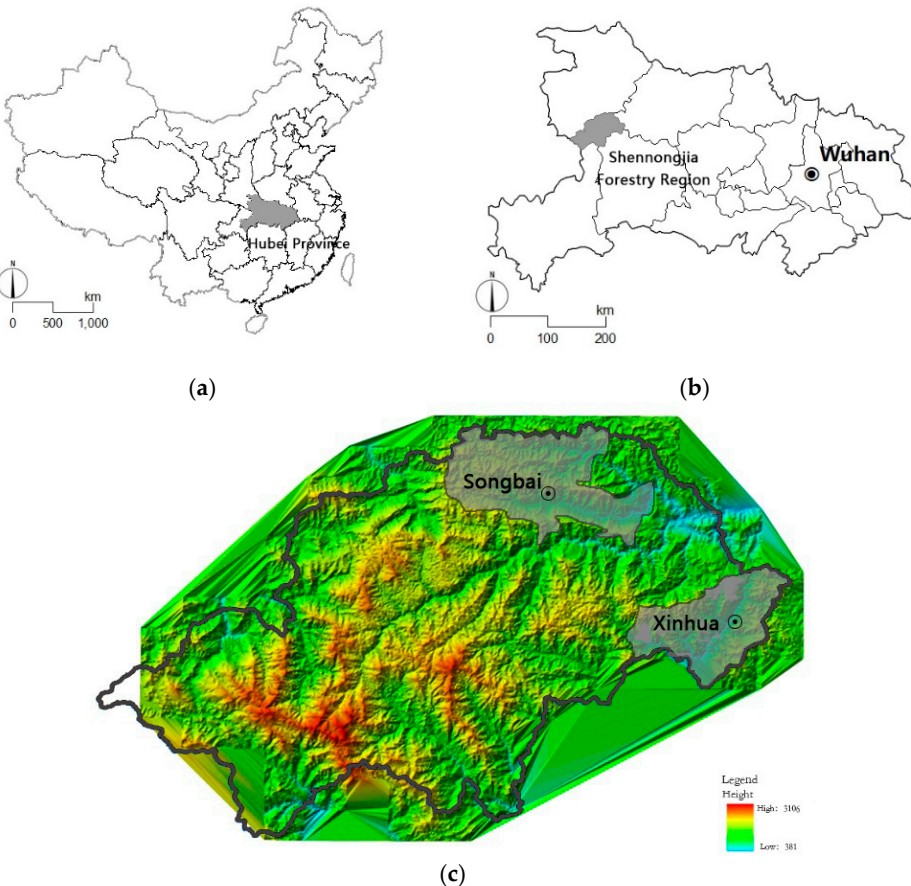

**Figure 3.** Location map: (**a**) Location of Hubei province in China; (**b**) location of Shennongjia in Hubei province; (**c**) location of Songbai town and Xinhua town in Shennongjia Forestry District.

Xinhua was chosen as a demo site to represent ordinary towns in SFD. The town of Xinhua is located at the southeastern border of SFD (Figure 3), with 207 square kilometers and a total population of 3826 in 2017. It is one of the most important border trade towns in this region. With the construction of the high-speed railway line from Zhenzhou to Chongqing, Xinhua is the only town with a high-speed

railway station in the district. As can be seen, it is stepping into a vital and important period of development at present. There will be more opportunities of construction activities as expected, therefore, harmonized spatial plans will play an important role in guiding its construction in the future.

Our team collected the spatial plans that have been adopted or in the preparation stage. Some plans are at the district level of SFD, including the 13th five-year plan for economic and social development, comprehensive strategic plan for urban and rural development (2016–2030), land use plan (2017–2030), the 13th five-year plan for forestry development, tourism development plan (2011–2030), and the 13th five-year plan for eco-environmental protection plan, and the plans in other relevant departments. Many other plans are at the level of towns and townships. In most cases, many towns and townships generally have their own urban and rural master plan and construction plan, respectively. Construction plan is a kind of detailed plan to serve as base reference. We collected all these plans from the towns and townships under the jurisdiction of SFD.

To ensure successful implementation of CDEW4MPI in SFD, the project team organized a series of workshops to train relevant personnel, including staff working in the government agencies and planning management technicians at the district level and township level.

## 4. CDEW4MPI

### 4.1. System Functions

Through the field investigation, interviews, and discussion with the local people, we found that the requirement of urban and rural construction management is very complicated. It is urgent to help them to identify what and where the existing and potential conflicts are among multiple plans. One of the main functions needed is to indicate the spatial difference among multiple plans. Considering most small cities and towns are located near ecological protected areas and farmland protection areas, the bottom-line control requirements are often used and should be meted by another main function. In the meantime, the professional and technical talents from remote support could be the major source for small cities and towns to carry out the smart management of various construction activities in the actual practice. Therefore, the system core of CDEW4MPI emphasizes the realization of two main functional modules, i.e., conflict-detecting and early-warning and offers an interface for users to perform them under the internet environment (Figure 1).

Elements for the conflict-detecting module include spatial boundaries, standards, and development quotas among the urban and rural development plan, land use plan, tourism plan, and forestry plan, etc. It can realize pairwise detection and analyze and report land use conflicts between any two plans after standardization. The early-warning module analyzes and reports potential encroachment of proposed local development plans on areas within basic control lines. Combing the practical needs of spatial governance in small cities and towns, we identify three types of basic control lines as early warning bottom lines for planning management, i.e., growth boundary line of urban and rural construction land, baseline for ecological protection, and red line for permanent farmland preservation.

The realization of these two models is accompanied by other functions of planning data retrieving, collaborative intelligent evaluation, threshold setting, result displaying, and statistics conducting. All these functions make the system operation simpler and more convenient to derive useful information from data access and result analysis in order to provide technical support for planning management.

### 4.2. System Operation Process

Figure 4 illustrates the operation process of CDEW4MPI supported by the cloud service platform. The users scattered in different places can access the datasets, software on the cloud platform in the internet environment. The system core offers interface for users to perform conflict-detecting and early-warning analyses. By supporting data retrieval and upload in the cloud, CDEW4MPI can realize the integration of planning, construction, and management data on the platform. Worthy of special note is that two types of users are designed in the user login module. One is the administrator user

(manager), who has the authority to manage, add, and delete ordinary users, and upload the basic data for early warning. User management interface is offered only for the operation of managers. The other is the ordinary user, who needs to apply for account number to login system. System interfaces can provide ordinary users to modify password, upload conflict detecting elements and data, conduct the conflict-detecting function, and the early-warning function.

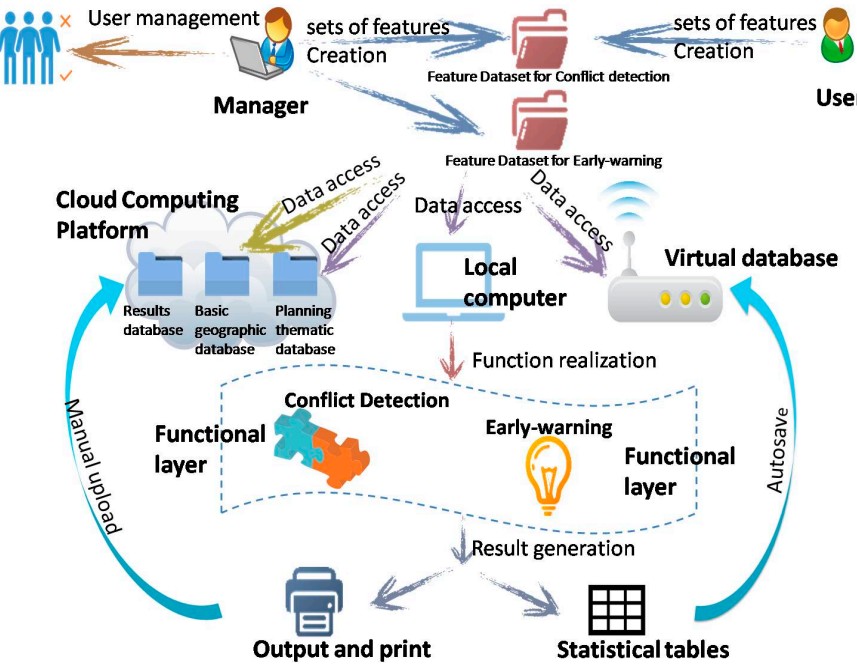

**Figure 4.** Operation flowchart on CDEW4MPI.

### 4.3. Planning Data Analysis

In order to promote economic and social development in SFD, different departments at the district level have worked out various plans successively. We can see that these plans come from different departments and they are quite different either in planning period or planning focus (Table 2). All these plans at the district level of Shennongjia are exerting a great influence on the development and construction in the towns under the jurisdiction by the hierarchical way.

**Table 2.** Sources and planning period of different plans from different departments in Shennongjia Forestry District (SFD).

| | Source | Beginning Year | End Year | Planning Period |
|---|---|---|---|---|
| 13th Five-year plan for economic and social development | Development and reform department | 2015 | 2020 | 5 years |
| Comprehensive strategic plan for urban and rural development | Urban and rural construction department | 2013 | 2030 | 18 years |
| Land Use Plan | Land management department | 2013 (revised in 2017) | 2030 | |
| 13th Five-year plan for Forestry development | Forestry department | 2015 | 2020 | 5 years |
| Tourism development plan | Tourism department | 2011 | 2030 | 20 years |
| 13th Five-year plan for Eco-environmental protection plan | Environmental protection department | 2015 | 2020 | 5 years |
| Spatial planning for Shennongjia Forestry District (draft) | County government | 2016 | 2030 | 15 years |

For SFD, there has been a new planning type that surfaced in recent years, i.e., spatial planning (Table 2). According to the spatial planning for SFD in 2016 (draft), three basic control lines (i.e., growth boundary line for urban and rural construction land, baseline for ecological protection, and red line for permanent farmland preservation) are designated. It is the first time this job has been done in this district.

In the Shennongjia Forestry District, with its own ecological vulnerability and limited land resource for construction development, spatial governance is becoming important and attaining close attention from all levels of governments. To serve both protection and development purposes, three lines have been delimited: Eco-environmental protection line, permanent preservation line for basic farmland, and urban construction land growth boundary. However, without the support of digital technology, it is rather difficult to implement the effective management and operation. The planning results are presented in text form, but the database and platform for actual management are missing. In this demonstration application, we unified the data of delimited lines into the cloud service platform and built up the spatial database of bottom lines for early-warning.

For Songbai and Xinhua, both towns have formulated urban and rural master planning in recent years for the detailed land use development (Table 3). It should be mentioned that land use planning in SFD covers the whole area at the district level and is also the basis for the land use management at the township level. Due to the different sources of plan-making, the spatial contradiction in the reality of the construction process is inescapable. Thus, both planning data in both towns are used for the demonstration application.

**Table 3.** Planning period of urban and rural master plan in Songbai and Xinhua.

| | Beginning Year | End Year | Planning Period |
|---|---|---|---|
| Master planning in Songbai town | 2013 | 2030 | 2013–2015 (short term) 2016–2020 (medium-term) 2021–2030 (long-term) |
| Master planning in Xinhua town | 2017 | 2030 | 2017–2022 (short-term) 2022–2030 (long-term) |
| Land use planning in Shennongjia Forestry District (which covered Songbai and Xinhua) | 2013 (revised in 2017) | 2030 | 2017–2020 (Short-term) 2020–2030 (long-term) |

As mentioned above, in Songbai and Xinhua, there are two important plans concerning space issues, i.e., the urban and rural master plan and land use plan, which respectively come from the urban and rural construction department and the land use management department. The spatial contradiction of both plans is most prominent. Accordingly, we select the two types of plans and use the conflict-detecting function of CDEW4MPI to test the detailed contradiction between both plans.

*4.4. System Application Demonstration*

The demonstration application of CDEW4MPI was finally implemented in Songbai and Xinhua in SFD. For both towns, the biggest spatial contradiction exists in urban and rural master plan and land use plan. At the same time, only the planning data of both towns can be relatively more easily obtained respectively through digital processing at this moment. On this basis, taking advantage of the cloud service planform of SFD supported and built by the Geo-data Center in Hubei province, the planning result data of master plan and land use plan of both towns are unified into the platform through remote access for conflict detecting. Using the conflict-detecting function developed by CDEW4MPI, we can quickly and effectively identify the spatial contradictions of both plans.

In the meantime, the data of three basic control lines (i.e., growth boundary line for urban and rural construction land, baseline for ecological protection, and red line for permanent farmland preservation), determined by Spatial Planning in 2016 at the district level of SFD, is also integrated into the cloud

service planform. Using the early-warning detecting function of CDEW4MPI, we can easily obtain the warning points of any plan that breaks through the basic control lines. All the spatial data used for the demonstration application can be uploaded and invoked on the cloud service planform.

Through the interface (Figure 5), to find the link to the conflict-detecting function and early-warning function, CDEW4MPI can help carry out the spatial difference analysis and spatial warning analysis.

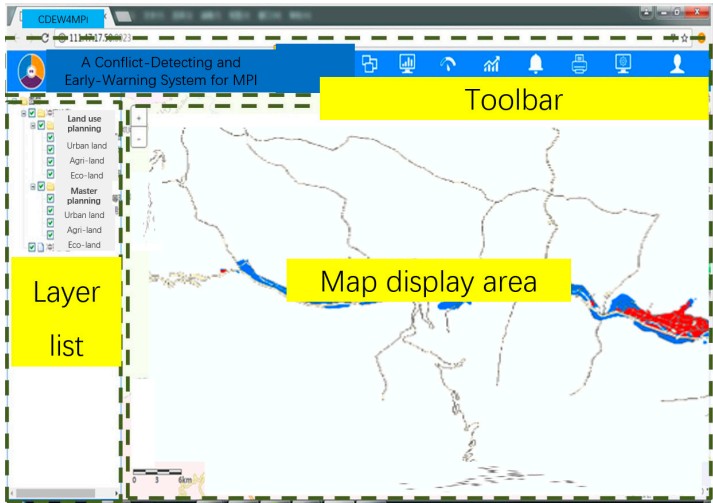

**Figure 5.** Interface of functions distribution in the main page.

## 4.5. Analysis Output for MPI Assessment

For all the results obtained by the CDEW4MPI operation in Songbai and Xinhua, the user can browse, locate, and view the attributes of each test. Moreover, all data can be transferred to the cloud service planform for storage, and for subsequent queries and modifications. The results can also be exported to the local computers for further analysis or printing.

Figure 6 displays some examples of the results of system operation. We can identify the distribution and calculate the size of the conflict areas. Then, the next step is to indicate the factors based on the basic geographic information database in the cloud service planform so as to find more targeted ways to coordinate the conflicts.

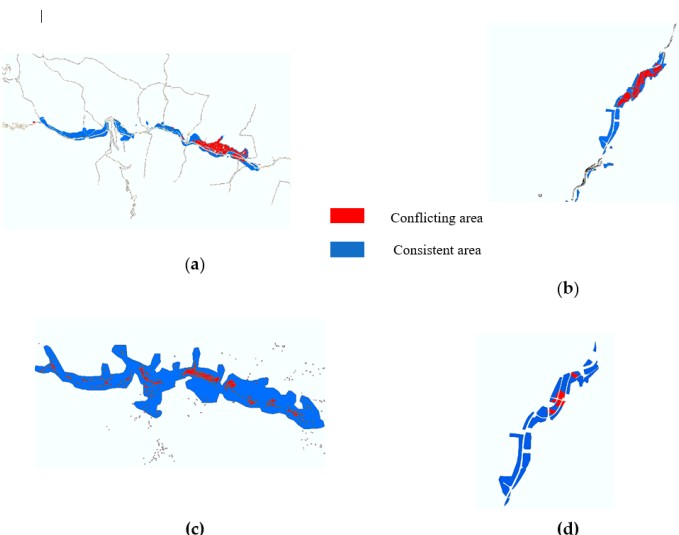

**Figure 6.** Results of the CDEW4MPI operation in Songbai town and Xinhua town in Shennongjia Forestry District: (**a**) Conflicting result of construction land use in the 'master plan' and 'land use plan' in Songbai town; (**b**) conflicting result of construction land use in the 'master plan' and 'land use plan' in Xinhua town; (**c**) early-warning for land use plan in village construction land breaking through the growth boundary of construction land in Songbai town; (**d**) early-warning for master plan in urban construction land breaking through the permanent preservation line of basic farmland in Xinhua town.

## 5. Discussion

Various agencies adopt different standards of land use, posing additional challenges to MPI. Like many other small cities and towns, planning output data from different departments possesses the characteristics of inconsistency in classification standards, coordinate systems, and data formats. The situation in Songbai and Xinhua is even worse due to the confusion management of data, weak technical foundations, and lack of professional expertise. In order to realize the basic data manipulation, the first step is to make all these data standardized and unify the coordinate system and data format before inputting the data into CDEW4MPI. In fact, our team took a significant amount of time to digitize data, match coordinate system, and create a unified planning database, which seems very basic work in the large cities, but is not basic work for these small cities and towns. However, from another point of view, this work also helps the local professional technicians and managers to realize the importance of digital data management. In addition, the unification of technical standards needs to be discussed and interviewed with the local people in order to adapt the realistic situation. Afterwards, all standardized data were integrated into the cloud service platform and the spatial database was built up before CDEW4MPI can be operated. Our research found that MPI for small cities and towns is very challenging and the system should be simple, practical, and easy to operate due to the lack of technical force and human resources.

Cloud service platforms provide the network environment of cloud computing through cloud server. In SFD, any kind of PC terminal can be connected to the home page of the public version of the portal network. By clicking the corresponding link, CDEW4MPI can be accessed remotely and all its functions can be carried out on the webpage. Local government and local people can save money and time from buying and managing servers, software, and datasets. All can be shared on the cloud platform. Due to the lack of professional and technical personnel, they are hunger of technical support and help from outside. The cloud computing platform provides the possibilities and seems very suitable for the development of the informatization in small cities and towns. The CDEW4MPI implementation offered the local managers and technicians to access the rich data from the remote location and obtain the visualized results about the spatial conflicts and problems of the plans. This increased access

motivated their great interests to learn more about the application of digital technology in spatial planning and management in the future.

It is very promising that in many small cities and towns or even rural areas in China, internet has been popularized to a great extent, making it possible and convenient for local people to browse online. From interacting with the local staff and citizen users, we found local people are willing to learn new things and eager to access the information outside. Application software for planning management based on cloud service platform is welcomed locally, but indeed it needs to be easily understood and operated without much difficult and complicated process. The experience of this system demonstration tells us that the goal of informatization construction for planning management should target the practical problem-solving and, what is more, to involve the participation of the local people.

It should also be noted that the CDEW4MPI based on provincial cloud-services is not free of challenges. The server for cloud-computing at the Hubei Provincial Geo-Information Center has quite a large capacity but still has a limit. Because of this capacity limit, the Center cannot keep all the projects the Center has served live online all the time. As an alternative service plan, the Center developed a switch schedule: The current, ongoing services with high priority stay active online. Other projects and services will be switched on in a demand-responsive mode. When the client, or Shennongjia in this case, needs to access CDEW4MPI, the Center will turn on CDEW4MPI, fully functional online. This arrangement is reasonable and understood by the local client because the provincial center provides the cloud services free of charge and there are too many projects for the Center to maintain live operation at the same time.

## 6. Conclusions

To achieve MPI objectives, small cities and towns face desperate shortage of multi-sourced database, technical knowhow, and professional expertise. Cloud computing technology enables end users to access software, data, and applications from distant locations. For small cities and towns, especially those located in remote areas, such technology makes it possible for them to access the huge amount of information and technical support in the advanced area with strong and abundant technology force. CDEW4MPI is such a system that proves to be functioning well in support of Shennongjia Forestry District, a municipality in the remote, mountainous region of central China to carry out MPI tasks.

CDEW4MPI at this phase of development remains relatively simple. It mainly operates two application functions for MPI, conflict detection and early warning detection. It is rather limited in that it cannot automatically offer adjustment solutions to the conflicts detected from different plans. Further refinements are necessary to incorporate additional analytical tools and coordinating mechanism for MPI and other policy and management purposes.

**Author Contributions:** Conceptualization, N.D. and M.Z.; data curation, N.D., M.Z. and J.H.; formal analysis, N.D.; funding acquisition, M.Z.; investigation, N.D., M.Z. and G.W.; methodology, N.D., M.Z. and J.H.; project administration, M.Z.; resources, N.D., M.Z. and G.W.; software, N.D. and J.H.; supervision, M.Z., G.W.; validation, N.D., M.Z. and J.H.; visualization, N.D. and J.H.; writing—original draft, N.D.; writing—review and editing, M.Z. and N.D.

**Funding:** This research was funded by China Ministry of Science and Technology through the 12th Five Year National Tech Support Grant for Smart Urbanization, 2015BAJ05B00.

**Conflicts of Interest:** The authors declare no conflict of interest.

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
