# Peer review of "A Conflict-Detecting and Early-Warning System for Multi-Plan Integration in Small Cities and Towns Based on Cloud Service Platform"

_smartcities, doi:10.3390/smartcities2030024_

Round 1

Reviewer 1 Report

The manuscript tackles an important problem of inconsistency and spatial conflicts between the plans prepared by various agencies. The authors propose a system (CDEW4MPI) which is to visualize the inconsistencies and this way help to overcome difficulties in digital data management and spatial coordination between planning documents.

The authors have addressed an important problem with an interesting experiment. The results are interpreted appropriately. However, the way the article is written does not provide sufficient information to give a sense of scientific soundness. 

The structure of the manuscript does not follow the customary, legible construction of a scientific paper. It lacks the division into sections such as ‘method’, ‘results’ and ‘discussion’. In effect, the hypotheses and speculations are not clearly identified so even if the problem described is important and the solution seems to be original, the results provide an advance more in current local practice than in the knowledge in the broader field of the application of digital technology in spatial planning and management.

In this manuscript, even if the results are interpreted appropriately, their significance might be questioned because of a low number of applications described. The paper mentions the application in only two locations in one district. For more representative outcomes, it would be preferable to investigate some broader choice of applications. 

The bibliography is very brief and seems to be focused on a narrow area of the project without usually required a broader background and deeper insight into a specific area of knowledge.

The English language of the paper is understandable although needs some minor improvements and proofreading.

Thanks to the significance of its topic, the paper could attract a wide readership but its lecture might not provide sufficient advance towards the current knowledge. Authors are aware that the described software is at its early phase of development and some refinements of it are necessary to incorporate more analytical tools and coordinating mechanism to meet policy and management purposes. 

In my opinion, the article might contribute some insight to the possible improvement of the practical tools for local planning if it was rewritten in a way to show a proper research design and present results more clearly, considering also the wider context of CDEW4MPI application.

Specific comments:

In verse 232, the is a link: http://111.47.17.49:5515/portal-web-public/home - which is to provide access to CDEW4MPI and all its functions but actually, it does not work.

Reviewer 2 Report

At first glance, this is an interesting paper and topic.

Upon closer inspection it does not contain enough substance nor understanding of planning, plan making and the details that go into the assessment of "conflicts."  

It's not clear how subjective information is managed and included.

The figures and discussion of the system content doesn't show how the evaluation is conducted nor does it adequately capture the diversity of variables, factors, conditions, constraints, requirements involved in planning.

There are other conflict or consistency or agreement studies and approaches for handling this.

Insufficient of the context both of the case studies as well as the system and the planning requirements. 

Without more details and more connection to the international planning literature, the paper is of limited value.  The literature review is deficient.  

The problems will not be solved by making this cloud based. 

Processes and decision rules and criteria for reaching agreement need to clarified.

It's unclear how engagement and consideration of values, interests, stakeholders, residents, and community concerns are included.   

Round 2

Reviewer 1 Report

The paper has been profoundly improved. In present form, it is sufficiently robust to be published. Proofreading and minor stylistic improvement by a native speaker would be welcome, however.